# Association of Plasma Placental Growth Factor with White Matter Hyperintensities in Alzheimer’s Disease

**DOI:** 10.3390/biom15101367

**Published:** 2025-09-26

**Authors:** Kazuya Igarashi, Tamao Tsukie, Kazuo Washiyama, Kiyoshi Onda, Yuki Miyagi, Shoya Inagawa, Soichiro Shimizu, Akinori Miyashita, Osamu Onodera, Takeshi Ikeuchi, Kensaku Kasuga

**Affiliations:** 1Department of Molecular Genetics, Brain Research Institute, Niigata University, Niigata 951-8585, Japan; igarashikazuya0711@gmail.com (K.I.); tsukie@bri.niigata-u.ac.jp (T.T.); miyashi2020.bri@niigata-u.ac.jp (A.M.); 2Department of Neurology, Brain Research Institute, Niigata University, Niigata 951-8585, Japan; onodera@bri.niigata-u.ac.jp; 3Department of Neurosurgery, Niigata Rinko Hospital, Niigata 950-8725, Japan; gctdementia@gmail.com; 4Department of Neurosurgery, Niigata Neurosurgical Hospital, Niigata 950-1101, Japan; kiyoshionda23@gmail.com; 5Department of Geriatric Medicine, Tokyo Medical University, Tokyo 160-8402, Japan; yuki8502@tokyo-med.ac.jp (Y.M.); inagawa.shoya.8u@tokyo-med.ac.jp (S.I.); soichiro@tokyo-med.ac.jp (S.S.); 6Department of Diagnostic Innovation Science, National Center for Geriatrics and Gerontology, Obu 474-8511, Japan

**Keywords:** Alzheimer’s disease, white matter hyperintensity, plasma biomarker, placental growth factor, PlGF

## Abstract

Autopsy studies have shown that Alzheimer’s disease (AD) often coexists with cerebrovascular injury, affecting cognitive outcomes and the effectiveness of anti-amyloid-beta (Aβ) drugs. No fluid biomarkers of cerebrovascular injury have been identified yet. We investigated the association between white matter hyperintensities (WMH) severity and fluid biomarkers, including cerebrospinal fluid (CSF) neurofilament light chain and plasma placental growth factor (PlGF) levels. This study included 242 patients from memory clinics. Magnetic resonance imaging (MRI), CSF, and plasma samples were collected. Patients were classified as AD+ or non-AD based on the CSF Aβ42/Aβ40 ratio. In the discovery cohort (79 AD+ and 20 non-AD patients with 3D-T1 images), we analyzed the association between WMH volume and plasma PlGF. In the validation cohort (54 AD+ patients without 3D-T1 images), we analyzed the association between WMH grading and plasma PlGF. Among AD+ patients in the discovery cohort, plasma PlGF levels remained significantly associated with WMH volume and grading after adjusting for age, sex, and global cognition. Among the AD+ patients in the validation cohort, the high-PlGF (above median) group had significantly greater WMH volumes and a higher number of patients with a high WMH grading than the low-PlGF (below median) group. Plasma PlGF is a promising marker of cerebrovascular injury in AD.

## 1. Introduction

The prevalence of dementia is increasing worldwide, with Alzheimer’s disease (AD) being the leading cause. With the advent of anti-β-amyloid (Aβ) antibody drugs in clinical practice, precise in vivo diagnosis of AD has become more crucial than ever before [1,2]. According to the revised criteria proposed by the Alzheimer’s Association in 2024, individuals with a decreased cerebrospinal fluid (CSF) Aβ42/Aβ40 ratio, the core biomarker, are diagnosed with AD [3].

Autopsy studies have revealed that AD is often accompanied by co-pathologies, such as cerebrovascular injury, which is becoming more prevalent with aging [4,5,6,7]. Some studies have reported that cerebrovascular lesions comorbid with AD contribute to cognitive decline [6,8,9], whereas other studies have suggested that they are not associated with cognitive decline or even slow its progression [10,11,12]. However, cerebrovascular injury may affect the effectiveness of anti-Aβ antibody drugs; thus, biomarkers that can detect coexisting vascular injury pathology in AD need to be established [3,13].

White matter hyperintensities (WMH) on brain magnetic resonance imaging (MRI) mainly reflect vascular brain injury, particularly small vessel disease [14,15,16]. Therefore, in the revised criteria, WMH on MRI is listed as a co-pathological marker of vascular brain injury in AD [3]. However, fluid biomarkers that reflect vascular brain injury are yet to be established. Elevated CSF or plasma levels of neurofilament light chain (NfL), a non-specific marker of neurodegeneration, reflect vascular injury [17] and have been associated with WMH in patients with AD [18,19,20]. Although many aspects of small vessel disease remain unclear, they are known to disrupt the blood–brain barrier (BBB) due to damage to endothelial cells and pericytes [21]. Placental growth factor (PlGF), originally identified in the placenta, is a member of the vascular endothelial growth factor family and is expressed in cerebral vascular endothelial cells. It plays an important role in angiogenesis and inflammation, making it a promising biomarker for vascular pathologies [22]. Several studies have examined CSF PlGF levels in relation to white matter pathology. In the Swedish BioFINDER study, higher CSF PlGF levels have been reported to be associated with greater WMH volumes, independent of Aβ status, in individuals without dementia [23]. In a Chinese AD continuum cohort, CSF PlGF was correlated with BBB permeability and WMH volume [24]. Studies investigating plasma and serum PlGF levels have yielded promising results. Elevated plasma levels of PlGF in patients with vascular cognitive impairment have been associated with the severity of white matter lesions and cognitive decline [25,26]. Recently, a study reported that elevated serum PlGF levels were associated with a higher WMH burden in patients with AD [27]. However, in a post-minor stroke cohort, serum PlGF levels were associated with BBB permeability, but not with WML volume [28]. These findings suggest that the relationship between PlGF and white matter lesions may be context-dependent or disease-specific, warranting further investigation.

Thus, this study aimed to analyze the association between WMH volume on MRI and CSF NfL and plasma PlGF levels in a memory clinic cohort of patients with and without AD. The association between WMH grading and plasma PlGF levels was validated in another cohort of patients with AD. Our data suggest that plasma PlGF is a promising marker of vascular injury in AD.

## 2. Materials and Methods

### 2.1. Participants

We included 242 sequential patients who visited the memory clinics at Niigata University and related institutions for diagnostic purposes between September 2016 and August 2024 and had available head MRI fluid-attenuated inversion recovery (FLAIR) images, CSF samples, and plasma samples (Figure 1). Patients were grouped according to the availability of 3D-T1 images, which are necessary for the calculation of WMH volume.

Of the 107 patients with 3D-T1 images, 3 patients with WMH volume analytic error, 3 patients with low quality (intra-coefficient of variation > 20%) of CSF NfL, and 2 patients with outliers of plasma PlGF were excluded. The remaining 99 patients were included in the discovery cohort, of whom 79 with a clinical diagnosis of MCI [29] or probable AD dementia [30] and a positive CSF Aβ42/Aβ40 ratio (details are described below) were classified as AD+, and 20 with a negative CSF Aβ42/Aβ40 ratio were classified as non-AD, regardless of the clinical diagnosis (Figure 1). Non-AD patients included clinically diagnosed cases of 4 AD, 1 dementia with Lewy bodies, 4 frontotemporal dementia, and 11 idiopathic normal pressure hydrocephalus (iNPH). The clinical diagnostic criteria used in this study have been described in our previous report [31].

Of the 135 patients without 3D-T1 images, 81 patients with a negative CSF Aβ42/Aβ40 ratio, regardless of clinical diagnosis, were excluded, and 54 AD+ patients were included in the validation cohort (Figure 1).

We evaluated dementia severity using the Clinical Dementia Rating and assessed global cognition using the Mini-Mental State Examination (MMSE). This study was conducted in accordance with the Declaration of Helsinki and was approved by the Ethics Committee of Niigata University (Approval No. 2019-0239). Written consent for participation was obtained from all patients or their proxies.

### 2.2. CSF Collection and Analysis

CSF samples were collected via lumbar puncture at each institution and sent to Niigata University. The samples were aliquoted at a volume of 0.5 mL and then stored at −80 °C until the assay. CSF concentrations of Aβ42 and Aβ40 were measured using the V-PLEX Aβ Peptide Panel 1 (6E10) (Meso Scale Discovery, Rockville, MD, USA), and the Aβ42/Aβ40 ratio was calculated. Aβ42/Aβ40 ratios less than 0.072 were considered positive [31]. The CSF concentrations of phosphorylated tau (p-tau) were measured using INNOTEST PHOSPHO-TAU (181P) (Fujirebio, Ghent, Belgium) and converted to the measurement values used by the AlzBio3 kit (Fujirebio, Ghent, Belgium) as previously explained [31]. The CSF concentrations of NfL were measured using the R-PLEX Human Neurofilament L Antibody Set (Meso Scale Discovery, Rockville, MD, USA). All analyses were conducted in duplicate by experienced technical assistants. The intra- and inter-assay coefficients of variation were <20% for all assays.

### 2.3. Plasma Collection and Analysis

Blood samples were obtained by venipuncture into tubes containing ethylenediaminetetraacetic acid and centrifuged at 2000× *g* for 15 min at 4 °C. Plasma was extracted and aliquoted at a volume of 0.5 mL and then stored at −80 ℃ until the assay. Plasma concentrations of PlGF were measured using the V-PLEX Plus Human PlGF Kit (Meso Scale Discovery, Rockville, MD, USA). All analyses were conducted in duplicate, and the intra- and inter-assay coefficients of variation were <20% for all assays.

### 2.4. Magnetic Resonance Imaging (MRI)

FLAIR images were obtained for all cases. Cases in which 3D-T1 images were obtained simultaneously were considered as the discovery cohort (Figure 1). The details of the MRI parameters at each facility are listed in Appendix A.

WMH volumes were calculated using 2D FLAIR and 3D T1 images with Brain Anatomical Analysis using diffeomorphic deformation (https://www.shiga-med.ac.jp/hqbioph/iBaad/page0.html, accessed on 16 January 2025), a certified medical device in Japan.

In the discovery cohort, deep and subcortical WMH (DSWMH) were visually graded by each attending physician (dementia specialist) and a neurologist (KI) in accordance with the guidelines adopted by the Japan Brain Dock Society [32], with grades ranging from 0 to 4, with higher scores representing more severe lesions. A score of 0 indicated absence; 1 indicated enlarged perivascular space, including (1) diameter ≤3 mm, boundary sharp, or (2) any evidence suggesting enlarged perivascular space; 2 indicated punctate or discrete foci on the subcortical and deep white matter; 3 indicated confluent foci on the deep white matter; and 4 indicated confluence that was widely distributed on most of the white matter. The grades were concordant in 74 of the 99 patients (74.7%). If the initial grading was discordant, another dementia specialist/neurologist (KK) provided the grading system. If KK’s grading matched that of either the attending physician or the KI, that grade was used (18/25, 72.0%). If all three had different grades (n = 7), consensus was reached through discussion. In the validation cohort, DSWMH was visually graded by each attending physician in accordance with the grading system mentioned above.

### 2.5. Statistical Analysis

Statistical analyses were performed using GraphPad Prism software version 10.4.1 (GraphPad Software Inc., La Jolla, CA, USA). Group comparisons for continuous variables, such as age and MMSE scores, were performed using the Mann–Whitney U test for two-group comparisons and the Kruskal–Wallis test for comparisons involving three groups. Simple linear regression analysis was performed to assess the association between continuous variables, such as biomarker levels and WMH volume. Group comparisons of categorical data were performed using the χ^2^ test. Correlations between the two datasets were identified using Spearman’s rank correlation coefficient. In the discovery cohort, the association between WMH volume and biomarkers was analyzed using linear multiple regression analysis. WMH volume was designated as the objective variable across all models. For explanatory variables, we used either CSF NfL or plasma PlGF alone (Model 1), followed by the sequential addition of age and sex (Model 2), MMSE score (Model 3), and CSF biomarkers (Aβ42 in Model 4, Aβ42/Aβ40 ratio in Model 5, and p-tau181 in Model 6). In Model 7, CSF NfL and plasma PlGF levels were combined to examine their association with WMH volume.

In the validation cohort, the association between DSWMH severity and plasma PlGF (dichotomized by median) was evaluated using multivariable logistic regression models. Age, sex, MMSE score, body mass index (BMI), and serum creatinine level were used as covariates. In Model 1, DSWMH severity (low vs. high) was the dependent variable, and plasma PlGF was the independent variable. In Model 2, age and sex were included as covariates. In Model 3, the MMSE score was added as a covariate. In Model 4, BMI and serum creatinine levels were added as covariates.

For all analyses, *p*-values < 0.05 were considered statistically significant.

## 3. Results

In the discovery cohort comprising patients with available WMH volume, 99 patients were included, of whom 79 were classified as AD+ based on clinical diagnosis and a positive CSF Aβ42/Aβ40 ratio. Twenty non-AD patients were negative for the CSF Aβ42/Aβ40 ratio, irrespective of clinical diagnosis. The demographic characteristics and biomarkers of the patients in the discovery cohort are presented in Table 1. Across the entire study sample (n = 99), we observed a median age of 76 years with 58.6% female participants and a median MMSE score of 24. DSWMH grade was widely distributed between 0 and 4 in the discovery cohort. Our analysis incorporated key demographic and clinical characteristics that influence the WMH volume. Across the entire sample, we found that WMH volume was strongly correlated with age (*r* = 0.602, *p* < 0.001), confirming that age is a major determinant of white matter pathology. This strong correlation was maintained in the AD+ subgroup (*r* = 0.604, *p* < 0.001), suggesting a consistent effect of aging on white matter integrity, regardless of the AD pathology status. With regard to sex differences, we found that WMH volume was not significantly different between females and males in the whole sample (*p* = 0.437) or in the AD+ subgroup (*p* = 0.675), indicating that sex may not be a major determinant of white matter lesion burden in our study population. Regarding clinical characteristics, MMSE scores in the whole cohort exhibited a trend toward a negative correlation with WMH volume; however, this association was not statistically significant (*r* = −0.172, *p* = 0.089). In contrast, in the AD+ subgroup, MMSE scores were weakly but significantly negatively correlated with WMH volume (*r* = −0.224, *p* = 0.047). This suggests that white matter lesions may have a more pronounced impact on cognitive function in individuals with AD pathology than in those without.

In all samples, linear multiple regression analysis showed that CSF NfL levels were significantly associated with WMH volume. The association remained significant after adjusting for age, sex, and MMSE score (Table 2, Models 1–3). Furthermore, given that CSF Aβ42 has been reported to be associated with vascular injury imaging markers [33], we included it as a covariate in our analysis (Model 4); however, the results remained significant. Additionally, after adjusting for the CSF Aβ42/Aβ40 ratio and CSF p-tau181 (Models 5 and 6), which reflect AD pathology, the association remained significant. Interestingly, when plasma PlGF and CSF NfL levels were incorporated into the same model (Model 7), both showed significant independent associations with WML volume. Plasma PlGF was also significantly associated with WMH volume. However, this association was no longer significant after adjusting for age and sex (Table 2). In the discovery cohort, plasma PlGF levels were significantly correlated with age but showed no significant sex differences (Appendix A), suggesting that the observed association between PlGF and WMH volume may be influenced by age. Although CSF Aβ42/Aβ40 ratio and CSF p-tau181 were not significantly associated with WMH volume in the overall analysis, simple linear regression analysis revealed significant associations with WMH volume specifically in the AD+ group for CSF NfL and CSF p-tau181 (Appendix A).

In the subgroup analysis, which included only AD+ patients, CSF NfL was significantly associated with WMH volume. However, the association was no longer significant after adjusting for the MMSE score, age, and sex (Table 2). In contrast, plasma PlGF was significantly associated with WMH volume (Figure 2A) and remained significant after adjusting for age, sex, and MMSE scores (Table 2). The association between plasma PlGF and WMH volume remained significant after adjusting for Aβ42 and p-tau181 (Models 4 and 6), whereas the association after adjusting for the Aβ42/Aβ40 ratio was marginal (Model 5, *p* = 0.054). Furthermore, when analyzing CSF NfL and plasma PlGF levels in the same model, only plasma PlGF showed a significant association with WML volume (Model 7).

Therefore, to confirm whether plasma PlGF reflects WMH volume in the AD brain, we stratified the AD+ patients into two groups according to the PlGF median (6.5 pg/mL). WMH volume was significantly greater in the high-PlGF group than in the low-PlGF group (Figure 2B).

Subsequently, we validated the association between plasma PlGF and WMH volume in AD in another cohort, in which all patients were diagnosed with AD+ by clinical diagnosis and CSF Aβ42/Aβ40 ratio. In the validation cohort, WMH volumes were not obtained because of the unavailability of 3D-T1 imaging; however, BMI and Cre information were available. Thus, the sample more closely resembles clinical practice. Considering that DSWMH grades were strongly correlated with WMH volume in the discovery cohort (Appendix A), we used DSWMH as a proxy for WMH volume in the validation cohort.

The demographic characteristics and biomarkers of the patients in the validation cohort are presented in Table 3. No significant differences in all demographics and biomarkers were found between the low-DSWMH (grade ≤ 1) and high-DSWMH (grade ≥ 2) groups. An exception to this finding was the plasma PlGF level, which was significantly higher in the high-DSWMH group than in the low-DSWMH group (*p* = 0.038).

Multiple logistic regression analysis was performed to examine the association between plasma PlGF levels and DSWMH severity. The results are summarized in Table 4. The analysis revealed that high plasma PlGF levels (higher than the median) were significantly associated with high DSWMH severity (grade ≥ 2) (Model 1; OR 4.09, 95% CI 1.25–15.13, *p* = 0.025). The association remained significant when age and sex (Model 2; OR 3.97, 95% CI 1.16–15.36, *p* = 0.034), MMSE score (Model 3: OR 3.70, 95% CI 1.07–14.38, *p* = 0.045), and BMI and serum creatinine, which are known to affect plasma biomarkers (Model 4; OR 3.85, 95% CI 1.10–15.35, *p* = 0.042), were included as covariates. Notably, plasma PlGF levels showed different age-related patterns between cohorts: significant correlation with age in the discovery cohort but not in the validation cohort, with no sex differences observed in either cohort (Appendix A). In the validation cohort, plasma PlGF levels did not differ significantly across BMI categories (underweight, normal, overweight) (*p* = 0.133, Appendix A), supporting the inclusion of BMI as a covariate.

Finally, the AD+ patients in the validation cohort were stratified into two groups according to the PlGF median (6.8 pg/mL). Significantly more patients in the high-PlGF group had a DSWMH grade ≥ 2 than those in the low-PlGF group (*p* = 0.0418, Figure 2C).

## 4. Discussion

This study showed that plasma PlGF levels were associated with WMH severity on brain MRI in patients with AD. CSF NfL was considered a positive control, and plasma PlGF was associated with WMH volume in whole samples of the discovery cohort. However, in the AD+ group of the discovery cohort, only plasma PlGF remained associated with WMH volume, independent of cognitive function. Similarly, in the validation cohort consisting only of AD+ patients, plasma PlGF was associated with WMH severity even after adjusting for confounding factors, such as BMI and serum creatinine, which affect plasma protein concentration. In the discovery cohort, the high-PlGF group had significantly larger WMH volumes than the low-PlGF group. Similarly, in the validation cohort, the high-PlGF group had significantly more patients with higher DSWMH scores than the low-PlGF group. This finding suggests that plasma PlGF is a marker of WMH severity in AD.

Autopsy studies have revealed that AD is often accompanied by co-pathologies [13,34,35]. The presence of co-pathologies may impact clinical outcomes, which suggests that evaluating biomarkers reflecting these co-pathologies is also recommended in the revised criteria [3]. In AD, the incidence of cerebrovascular injury increases with age [4,5,7,36]. In the revised criteria, WMH on brain MRI is listed as a marker for cerebrovascular injury associated with AD [3]. Regarding the detection of AD pathology, the usefulness of plasma biomarkers, particularly p-tau217, has been widely reported [37,38,39]. However, plasma p-tau217 is unable to detect the presence or absence of co-pathology in AD [40,41]. Given the associated cost-effectiveness and scalability, establishing a blood biomarker to detect co-pathologies would enhance diagnostic efficiency when combined with p-tau217 in a single blood test. Therefore, using CSF NfL as a positive control, we evaluated the utility of PlGF, a promising blood biomarker reflecting small vessel disease and associated with WMH volume [25,26,27].

In the discovery cohort, we confirmed the previously reported association between CSF NfL and WMH volume [18,19,20]. Across the entire study sample, CSF NfL remained significantly associated with WMH volume, even after adjusting for AD pathology markers (CSF Aβ42/Aβ40 ratio and p-tau181), supporting CSF NfL as a marker of white matter damage independent of AD pathology. However, in the AD+ subgroup, this association disappeared after adjusting for MMSE, suggesting that CSF NfL may not effectively capture the white matter lesions that affect cognitive function in AD. In contrast, the plasma PlGF levels exhibited different patterns. In the full cohort, it was not significantly associated with the WMH volume after adjusting for age and sex. However, in the AD+ subgroup, the association remained significant, even after MMSE adjustment. This suggests that PlGF may be influenced more by age-related white matter changes under non-AD conditions. Indeed, in the whole-cohort analysis, when both plasma PlGF and CSF NfL were included in the same model, they were independently associated with WML volume, indicating that these two markers likely capture different aspects of white matter lesions. However, in the AD+ subgroup, when plasma PlGF and CSF NfL were incorporated into the same model, only PlGF was associated with the WMH volume. Studies in the United States have identified plasma PlGF as a marker of WMH in cerebrovascular cognitive impairment [25,26]. A recent study from Singapore [27] reported that elevated serum PlGF was associated with a higher WMH burden in patients with AD, which is consistent with our findings in a Japanese cohort. These studies across diverse ethnic backgrounds suggest a consistent relationship between PlGF and WMH that may be relevant across populations, supporting the potential role of PlGF as a biomarker of cerebrovascular pathology in AD. However, it is important to note that a recent study in a post-minor stroke cohort from the United Kingdom did not find an association between plasma PlGF levels and WMH volume [28]. This discrepancy suggests potential racial differences in the relationship between PlGF and WMH or, alternatively, that our findings may be specific to AD pathology. Future replication studies across diverse cohorts are necessary to clarify these possibilities and establish the generalizability of our findings.

Regarding why PlGF is more prominently associated with WMH volume in AD+ individuals, several neurobiological explanations can be proposed. WMH observed in AD primarily reflects cerebrovascular injury pathology, whereas it is considered to be related to Aβ pathology and/or tau pathology [14,15,16,29,42,43,44]. To disentangle these potential mechanisms, we investigated the association between plasma PlGF and WMH volume after adjusting for AD-related CSF biomarkers in the AD+ subgroup. Interestingly, plasma PlGF remained significantly associated with WMH volume even after adjusting for Aβ42, which typically decreases in the CSF along with Aβ40 in cerebral amyloid angiopathy (CAA) [45,46]. This suggests that white matter lesions associated with PlGF do not primarily reflect CAA-mediated ischemia. Instead, PlGF may reflect chronic ischemia caused by small vessel disease, which is commonly comorbid with AD pathology, leading to increased BBB permeability and contributing to the expansion of WMH volume.

Alternatively, the association between plasma PlGF and WMH volume became marginal after adjusting for CSF Aβ42/Aβ40 ratio but remained significant after p-tau181 adjustment. This differential pattern suggests that plasma PlGF elevation may partially reflect the white matter pathology associated with Aβ plaque deposition [14,44] rather than tau-related processes [15]. This interpretation is consistent with our observation that the association between PlGF and WMH was stronger in the AD+ group than in the whole cohort, which included non-AD conditions such as iNPH. One possibility is that PlGF increases in response to inflammation induced by Aβ deposition and subsequently contributes to the expansion of WMH volume through BBB disruption caused by inappropriate angiogenesis.

It remains challenging to conclusively determine whether PlGF-associated white matter lesions in the AD+ group are predominantly related to comorbid ischemic lesions or to the Aβ pathology itself. The weaker association between plasma PlGF and WMH volume observed in the whole group compared to the AD+ subgroup may be partly explained by our cohort composition, in which most non-AD patients were diagnosed with iNPH. Further investigations in cohorts with vascular cognitive impairment would be valuable to better characterize the relationship between plasma PlGF and WMH across different neurological conditions.

In the validation cohort, 20% of the patients in the high-PlGF group had low DSWMH. This finding suggests that PlGF can be used to identify the early stages of cerebrovascular injury. Diffusion tensor imaging can detect cerebrovascular injury lesions earlier than WMH [26,47,48]. Future studies should confirm the correlation between PlGF and cerebrovascular injury using diffusion tensor imaging to verify whether PlGF can serve as an early marker of cerebrovascular injury.

Some studies have reported that cerebrovascular injury coexisting with AD increases the risk of developing dementia [6,9,49,50,51], whereas other studies have suggested that it slows longitudinal cognitive decline [10,11,12]. With the same level of AD pathology, cognitive function worsens when cerebrovascular injury coexists [6,9]. However, with the same level of cognitive function, patients with AD coexisting with cerebrovascular injury are considered to have a milder AD pathology than those without cerebrovascular injury [7,52]. Given these considerations, patients with high plasma PlGF levels may have milder AD pathology. Therefore, thorough management of vascular risk may be effective in slowing disease progression. In addition, anti-Aβ antibodies are expected to be effective in slowing progression in cases with mild tau pathology [1]. In this context, along with cognitive examination and AD core biomarkers, stratifying with plasma PlGF may help identify groups where anti-Aβ antibodies are effective. In the future, stratifying cohorts using AD-related biomarkers, such as plasma p-tau217 and plasma PlGF, will be necessary. This approach will enable more precise comparisons of longitudinal changes in cognitive function and the therapeutic effects of anti-Aβ antibody drugs. This stratification approach may not only guide therapeutic decisions but also inform screening strategies in broader clinical settings. Furthermore, plasma PlGF could serve as an accessible screening tool in primary care settings where MRI is not readily available, potentially facilitating early identification of patients who would benefit from further neurological evaluation.

This study had some limitations. First, in the discovery cohort, the analysis results differed between the entire group and the AD+ group alone. However, in the non-AD groups, such as the iNPH group, the small number of cases may have resulted in a lack of significant differences. Future studies should analyze the correlation between plasma PlGF and WMH volume in patients without AD, particularly in those with vascular cognitive impairment. Second, information on vascular risk factors, such as hypertension, diabetes, cardiovascular disease, and other comorbidities, which are commonly observed in older adults, was not available in this study. Future studies should include comprehensive health data and medications to better characterize the participants and, more precisely, define the relationship between PlGF, vascular pathology, and AD. Third, this study did not confirm the extent of Aβ and tau pathology using positron emission tomography or the presence or absence of CAA using T2* or susceptibility-weighted imaging. Therefore, the impact of these lesions on the association between plasma PlGF and WMH in AD warrants further investigation. Fourth, the cross-sectional design of our study prevented us from establishing causal relationships between PlGF levels and WMH development or progression. Longitudinal studies are needed to determine whether elevated PlGF levels precede WMH formation or increase it as a consequence of existing vascular pathology. Finally, East Asian populations have a high burden of concomitant cerebrovascular diseases. Thus, validation in other racial and ethnic groups is required to ensure the generalizability of the findings.

## 5. Conclusions

In conclusion, plasma PlGF levels are associated with WMH severity, particularly in patients with AD. Plasma PlGF is a potential fluid marker of cerebrovascular injury in AD. Future studies should stratify AD using plasma PlGF levels and analyze their relationship with cognitive prognosis.

## Figures and Tables

**Figure 1 biomolecules-15-01367-f001:**
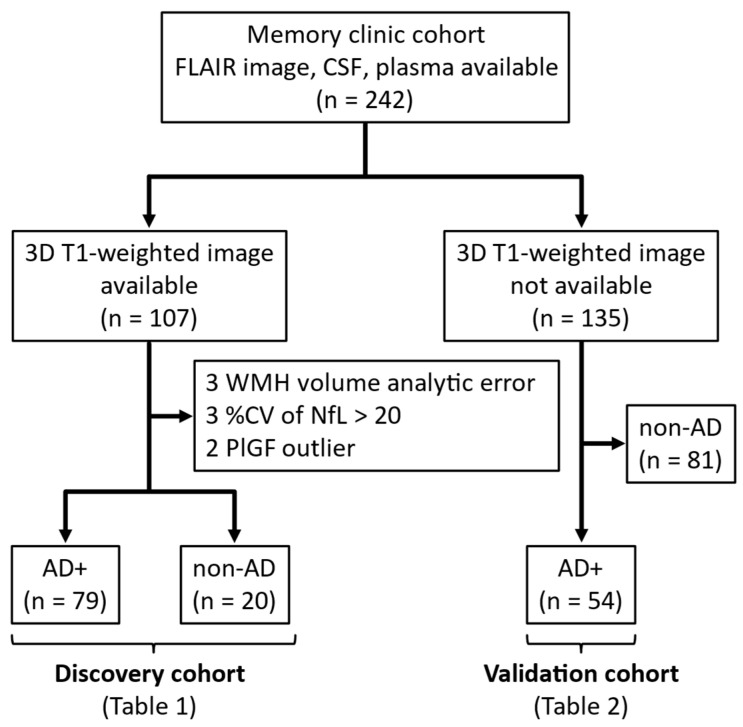
Selection process of participants for analysis. Abbreviations: AD, Alzheimer’s disease; CV, coefficient of variation; CSF, cerebrospinal fluid; FLAIR, fluid-attenuated inversion recovery; NfL, neurofilament light chain; PlGF, placental growth factor; WMH, white matter hyperintensity.

**Figure 2 biomolecules-15-01367-f002:**
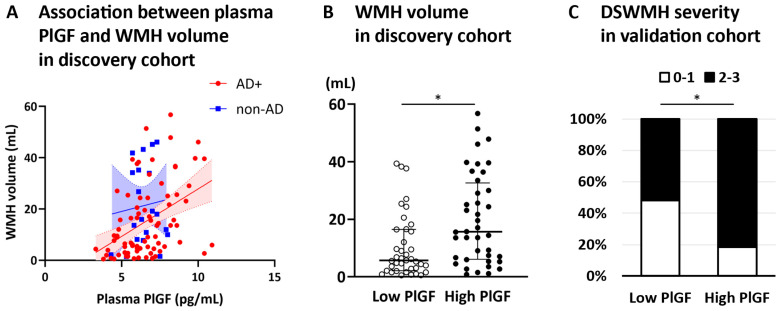
Plasma PlGF-WMH volume association and WMH severity comparison by PlGF levels. The scatter plot illustrates the relationship between plasma PlGF levels and WMH volume (**A**): AD+ (red) and non-AD (blue), the regression line (solid), and the 95% confidence intervals (dashed lines) are displayed. In the discovery cohort, the WMH volume was significantly greater in the high-PlGF group than in the low-PlGF group (Mann–Whitney U test, * *p* < 0.05) (**B**). In the validation cohort, significantly more patients in the high-PlGF group had DSWMH grades of 2 or higher than those in the low-PlGF group (Chi-square test, * *p* < 0.05) (**C**). Abbreviation: DSWMH, deep and subcortical white matter hyperintensity; PlGF, placental growth factor; WMH, white matter hyperintensity.

**Table 1 biomolecules-15-01367-t001:** Demographic characteristics and biomarkers of the discovery cohort.

	Whole	AD+	Non-AD
	n = 99	n = 79	n = 20
**Age, years**	76 (10)	75 (10)	80 (9)
**Female/Male**	58/41	50/29	8/12
**Clinical diagnosis**			
**AD**	83	79	4
**DLB**	1	0	1
**FTD**	4	0	4
**iNPH**	11	0	11
**CDR score**			
**0.5**	64	48	16
**1**	22	19	3
**2**	7	6	1
**3**	2	2	0
**NA**	4	4	0
**MMSE score**	24 (5)	24 (4)	24 (4)
**DSWMH grading**			
**0**	17	14	3
**1**	30	26	4
**2**	28	21	7
**3**	21	16	5
**4**	3	2	1
**WMH volume, mL**	12.0 (20.9)	9.3 (20.2)	17.0 (24.9)
**CSF Aβ42/Aβ40**	0.046 (0.018)	0.044 (0.015)	0.069 (0.028)
**CSF p-tau181, pg/mL**	50.1 (29.8)	56.7 (24.0)	25.0 (19.7)
**CSF NfL, pg/mL**	4501.6 (3115.2)	4501.6 (3069.6)	4733.4 (2870.6)
**Plasma PlGF, pg/mL**	6.4 (1.8)	6.5 (2.2)	6.4 (1.1)

Note: Continuous variables are presented as median (interquartile range), while categorical variables are presented as numbers. Abbreviations: AD, Alzheimer’s disease; CDR, clinical dementia rating; CSF, cerebrospinal fluid; DLB, dementia with Lewy bodies; DSWMH, deep and subcortical white matter hyperintensity; FTD, frontotemporal dementia; iNPH, idiopathic normal pressure hydrocephalus; MMSE, Mini-Mental State Examination; NA, not available; NfL, neurofilament light chain; PlGF, placental growth factor; WMH, white matter hyperintensity.

**Table 2 biomolecules-15-01367-t002:** Linear association between WMH volume and biomarkers.

	Model 1	Model 2	Model 3	Model 4	Model 5	Model 6	Model 7
**Whole**							
**NfL**	**5.74 × 10^−4^ (2.04 × 10^−4^ to 9.43 × 10^−4^)** ***p* = 0.003**	**4.60 × 10^−4^ (1.41 × 10^−4^ to 7.79 × 10^−4^)** ***p* = 0.005**	**3.98 × 10^−4^ (9.86 × 10^−5^ to 6.98 × 10^−4^)** ***p* = 0.010**	**3.55 × 10^−4^ (5.49 × 10^−5^ to 6.54 × 10^−4^)** ***p* = 0.021**	**4.04 × 10^−4^ (1.01 × 10^−4^ to 7.07 × 10^−4^)** ***p* = 0.010**	**3.73 × 10^−4^ (7.05 × 10^−4^ to 6.75 × 10^−4^)** ***p* = 0.016**	**4.28 × 10^−4^ (1.31 × 10^−4^ to 7.24 × 10^−4^)** ***p* = 0.005**
**Age**	**–**	**0.87 (0.60 to 1.14)** ***p* < 0.001**	**0.87 (0.62 to 1.13)** ***p* < 0.001**	**0.89 (0.64 to 1.15)** ***p* < 0.001**	**0.88 (0.62 to 1.14)** ***p* < 0.001**	**0.82 (0.55 to 1.09)** ***p* < 0.001**	**0.74 (0.46 to 1.02)** ***p* < 0.001**
**Sex (Male)**	–	1.60 (−3.34 to 6.55) *p* = 0.521	3.50 (−1.21 to 8.22) *p* = 0.144	3.57 (−1.09 to 8.23) *p* = 0.132	3.59 (−1.18 to 8.35) *p* = 0.139	3.22 (−1.51 to 7.95) *p* = 0.180	3.81 (−0.84 to 8.46) *p* = 0.107
**MMSE**	–	–	**−0.90 (−1.36 to −0.44)** ***p* < 0.001**	**−0.84 (−1.30 to −0.38)** ***p* < 0.001**	**−0.90 (−1.36 to −0.43)** ***p* = 0.010**	**−0.91 (−1.37 to −0.45)** ***p* < 0.001**	**−0.88 (−1.33 to −0.42)** ***p* < 0.001**
**Aβ42**	–	–	–	−0.03 (−0.06 to 0.00) *p* = 0.070	–	–	–
**Aβ ratio**	–	–	–	–	−24.3 (−171.2 to 122.5) *p* = 0.743	–	–
**p-tau**	–	–	–	–	–	−0.057 (−0.15 to 0.037) *p* = 0.232	–
**PlGF**	–	–	–	–	–	–	**1.66 (0.04 to 3.27)** ***p* = 0.045**
**PlGF**	**3.50 (1.71 to 5.29)** ***p* < 0.001**	1.56 (−0.23 to 3.36) *p* = 0.087	–	–	–	–	–
**Age**		**0.79 (0.48 to 1.10)** ***p* < 0.001**	–	–	–	–	–
**Sex (Male)**		3.21 (−1.81 to 8.23) *p* = 0.207	–	–	–	–	–
**AD+**							
**NfL**	**1.36 × 10^−3^ (2.22 × 10^−4^ to 2.50 × 10^−3^)** ***p* = 0.020**	**1.10 × 10^−3^ (1.27 × 10^−4^ to 2.06 × 10^−3^)** ***p* = 0.027**	6.78 × 10^−4^ (−2.38 × 10^−4^ to 1.59 × 10^−3^) *p* = 0.145	–	–	–	–
**Age**	–	**0.85 (0.56 to 1.15)** ***p* < 0.001**	**0.86 (0.59 to 1.13)** ***p* < 0.001**	–	–	–	–
**Sex (Male)**	–	2.99 (−2.55 to 8.52) *p* = 0.286	4.53 (−0.63 to 9.68) *p* = 0.084	–	–	–	–
**MMSE**	–	–	**−0.94 (−1.43 to −0.45)** ***p* < 0.001**	–	–	–	–
**PlGF**	**3.70 (1.94 to 5.47)** ***p* < 0.001**	**1.95 (0.14 to 3.76)** ***p* = 0.035**	**1.67 (0.02 to 3.31)** ***p* = 0.048**	**1.69 (0.10 to 3.28)** ***p* = 0.037**	1.62 (−0.03 to 3.28) *p* = 0.054	**1.71 (0.08 to 3.35)** ***p* = 0.040**	**1.67 (0.04 to 3.31)** ***p* = 0.045**
**Age**	–	**0.72 (0.38 to 1.05)** ***p* < 0.001**	**0.74 (0.43 to 1.04)** ***p* < 0.001**	**0.80 (0.50 to 1.10)** ***p* < 0.001**	**0.75 (0.45 to 1.06)** ***p* < 0.001**	**0.67 (0.35 to 0.98)** ***p* < 0.001**	**0.71 (0.41 to 1.02)** ***p* < 0.001**
**Sex (Male)**	–	3.09 (−2.46 to 8.64) *p* = 0.271	4.71 (−0.38 to 9.80) *p* = 0.069	**4.97 (0.05 to 9.88)** ***p* = 0.048**	4.76 (−0.35 to 9.87) *p* = 0.067	4.82 (−0.23 to 9.88) *p* = 0.061	4.61 (−0.43 to 9.66) *p* = 0.073
**MMSE**	–	–	**−0.99 (−1.46 to −0.52)** ***p* < 0.001**	**−0.92 (−1.38 to −0.46)** ***p* < 0.001**	**−1.00 (−1.48 to −0.53)** ***p* < 0.001**	**−0.98 (−1.45 to −0.51)** ***p* < 0.001**	**−0.90 (−1.38 to −0.42)** ***p* < 0.001**
**Aβ42**	–	–	–	**−0.05 (−0.09 to −0.01)** ***p* = 0.012**	–	–	–
**Aβ ratio**	–	–	–	–	−84.2 (−316.9 to 148.5) *p* = 0.473	–	–
**p-tau**	–	–	–	–	–	−0.08 (−0.18 to 0.03) *p* = 0.150	–
**NfL**	–	–	–	–	–	–	6.84 × 10^−4^ (−2.13 × 10^−4^ to 15.8 × 10^−4^) *p* = 0.133

The estimates (95% confidence intervals) and *p*-values are shown. **Bold** values indicate statistical significance. Model 1: no adjustment; Model 2: adjusted for age and sex; Model 3: adjusted for age, sex, and MMSE score. Abbreviations: AD, Alzheimer’s disease; MMSE, Mini-Mental State Examination; NfL, neurofilament light chain; PlGF. Placental growth factor; p-tau, phosphorylated tau; WMH, white matter hyperintensity.

**Table 3 biomolecules-15-01367-t003:** Demographic characteristics and biomarkers of the validation cohort.

	Whole	Low DSWMH	High DSWMH
	n = 54	n = 18	n = 36
**Age, years**	79 (9)	77 (8)	81 (9)
**Female/Male**	33/21	10/8	23/13
**BMI**	21.7 (4.7)	22.3 (4.9)	21.6 (4.4)
**CDR score**			
**0**	2	1	1
**0.5**	31	13	18
**1**	19	4	15
**2**	2	0	2
**MMSE score**	23 (5)	24 (4)	22 (5)
**DSWMH grading**			
**0**	1	1	0
**1**	17	17	0
**2**	23	0	23
**3**	13	0	13
**Serum Cre, mg/dL**	0.74 (0.22)	0.72 (0.20)	0.78 (0.21)
**CSF Aβ42/Aβ40**	0.044 (0.014)	0.051 (0.014)	0.044 (0.015)
**CSF p-tau181, pg/mL**	51.8 (31.3)	47.2 (32.1)	52.3 (31.2)
**CSF NfL, pg/mL ***	5067.1 (3891.0)	5036.4 (1481.4)	5436.0 (4572.9)
**Plasma PlGF, pg/mL**	6.8 (2.2)	5.9 (1.8)	7.2 (1.8) **

* Seven samples (1 low DSWMH, 6 high DSWMH) were excluded due to low quality of measurement. ** Plasma PlGF level was significantly higher in the High DSWMH group than in the Low DSWMH group (Mann–Whitney *U* test, *p* < 0.05). Note: Continuous variables are presented as median (interquartile range), while categorical variables are presented as numbers. Abbreviations: Aβ, amyloid-β; BMI, body mass index; CDR, clinical dementia rating; Cre, creatinine; CSF, cerebrospinal fluid; DSWMH, deep and subcortical white matter hyperintensity; MMSE, Mini-Mental State Examination; NfL, neurofilament light chain; PlGF, placental growth factor; p-tau, phosphorylated tau.

**Table 4 biomolecules-15-01367-t004:** Multivariable logistic regression models for odds of High DSWMH.

	Model 1	Model 2	Model 3	Model 4
PlGF (high)	**4.09 (1.25–15.13)** ***p* = 0.025**	**3.97 (1.16–15.36)** ***p* = 0.034**	**3.70 (1.07–14.38)** ***p* = 0.045**	**3.85 (1.10–15.35)** ***p* = 0.042**
Age	–	1.08 (0.99–1.19) *p* = 0.107	1.08 (0.98–1.19) *p* = 0.133	1.07 (0.98–1.19) *p* = 0.157
Sex (Male)	–	1.07 (0.30–4.20) *p* = 0.919	0.91 (0.23–3.74) *p* = 0.889	0.89 (0.18–4.50) *p* = 0.884
MMSE score	–	–	0.92 (0.76–1.09) *p* = 0.357	0.94 (0.77–1.14) *p* = 0.529
BMI	–	–	–	0.93 (0.77–1.13) *p* = 0.475
Creatinine	–	–	–	2.48 (0.06–166.4) *p* = 0.643

Odds ratios (95% confidence intervals) and *p*-values are shown. **Bold** indicates statistical significance. Model 1: no adjustment; Model 2: age- and sex-adjusted; Model 3: adjusted for age, sex, and MMSE score. Model 4: adjusted for age, sex, MMSE, BMI, and serum Cre level. Abbreviations: BMI, body mass index; DSWMH, deep and subcortical white matter hyperintensity; MMSE, Mini-Mental State Examination; PlGF. Placental growth factors.

## Data Availability

The dataset generated and analyzed during the current study is not publicly available due to privacy restrictions and ethical approval limitations, but is available from the corresponding author upon reasonable request.

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
