# Peer review of "Association of Plasma Placental Growth Factor with White Matter Hyperintensities in Alzheimer’s Disease"

_biomolecules, 2025, doi:10.3390/biom15101367_

Round 1

Reviewer 1 Report

Comments and Suggestions for Authors

With great interest, I read the work by Kazuya Igarashi et al., who submitted a compelling paper exploring a potential fluid biomarker that may reflect vascular co-pathology in Alzheimer’s disease (AD). The authors established a well-characterized cohort of patients with and without AD, with the AD diagnosis confirmed using objective cerebrospinal fluid (CSF) biomarkers (Aβ42/Aβ40 ratio).

Based on their findings, the authors propose placental growth factor (PlGF) as a promising candidate marker for cerebrovascular injury in AD. The methodology is presented in a clear and transparent manner, and the study's limitations are appropriately acknowledged in the discussion. Overall, the manuscript is highly interesting and provides valuable new insights into the field.

Here are a few suggestions to further improve the manuscript:

  1. The association between white matter hyperintensities (WMH) and vascular pathology is discussed, but it should be noted that WMHs may result from multiple etiologies. A brief discussion of this point would strengthen the interpretation of findings.
  2. The clinical implications of identifying a fluid biomarker such as PlGF should be emphasized more strongly. In particular, the potential advantages over current neuroimaging techniques (e.g., accessibility, cost-effectiveness, early detection) could be elaborated.
  3. Please clarify the magnetic field strength used in the MRI imaging protocol, as this can impact WMH assessment.
  4. Tables 1 and 3 would benefit from a clear indication of which measures reached statistical significance.
  5. The abbreviation “WMH” should be defined in the abstract.

Author Response

Please see the attached file “Response letter_Reviewer 1_24Sep2025.pdf” for our detailed reply to Reviewer 1.

Reviewer 2 Report

Comments and Suggestions for Authors

The article by Igarashi K. and colleagues investigated the relationship of well established cerebrospinal fluid (CSF) biomarkers for Alzheimer´s disease (AD) amyloid beta (Aβ42, Aβ40) and  phosphorylated Tau, with the CSF and plasma circulating vascular brain injury marker neurofilament light chain NfL, in addition to placental growth factor (PGF, PlGF). They utilized a cohort of 242 patients with and without AD in which data on cognitive function and white matter hypersintensities (MH) , a marker of vascular injury, were available. As the authors point out, PlGF levels had already been associated with AD pathogenesis but whether these are also associated with white matter lesions remained unclear. They find an association between the grade of WMH with plasma circulating PlGF and suggest it as a promising new marker for vascular injury in AD.

The patient cohort, exclusion and stratification criteria are clearly presented as a flow-chart in Figure 1. Collection and quantification of fluid CSF and plasma analytes is decently described and non-invasive MRI measures are listed summarized in supplementary. Statistical methods are well described and data are presented in tables taking several confounding factors- such as age, sex, BMI, serum creatinine and MMSE score into account.  

Age was strongly WMH volume was strongly correlated with age independent of sex. Trends of a negative correlation among MMSE and WMH were found- a finding which emphasizes the need of biofluid markers for characterization AD in addition to imaging. Accordingly, NfL levels correlated well with WMH volume and this correlation was independent of age, sex, and classical CSF-beta-amyloid biomarkers. In contrast, PlGF – WMH correlations were found to depend on age and sex. Subgroup analysis revealed that PlGF correlated with disease severity independent of sex and age.

Critizism/suggestion:

While these tables are certainly informative, figures as shown in Figure 2 A and B appear much more attractive to me and I encourage the authors to present NfL, PlGF data in age, sex – and if possible BMI subsets (i.e. underweight, normal, overweight). For compariblitly with earlier published results, such plots can be privided also for Aβ42, Aβ40, pTau.

In summary, it appeared that plasma PlGF captures white matter lesion to a better extent than CSF NfL. The authors discuss their results carefully and point out the limitations of their study. I recommend publication after considering my suggestion to prepare additional scatter and boxplots as mentioned above.

Author Response

Please see the attached file “Response letter_Reviewer 2_24Sep2025.pdf” for our detailed reply to Reviewer 2.
